# A Ku-Band Magnified Active Tx/Rx Multibeam Antenna Based on a Discrete Constrained Lens

**Gianfranco Ruggerini [1], Pasquale Giuseppe Nicolaci [2] and Giovanni Toso [3],***

1    ST4I Space Technologies for Innovation, 00131 Rome, Italy; gianfranco.ruggerini@st4in.com
2    TICRA, DK-1119 Copenhagen, Denmark; pgn@ticra.com
3    Antenna and Sub-Millimeter Waves Section, European Space Agency, ESTEC,
     2200 AG Noordwijk, The Netherlands
*    Correspondence: giovanni.toso@esa.int

**Abstract:** This paper presents the design, manufacturing, and testing of a Ku-band magnified active Tx/Rx multibeam antenna based on a discrete constrained lens. Multibeam antennas based on constrained lenses generate a significant number of beams on a large operative frequency bandwidth. The main novelty and challenge in this development are associated with the combined transmitting and receiving functionality of this active array. An innovative RF building blockchain working in dual-polarization and covering both the transmitting and receiving bandwidth is successfully implemented. In order to improve the accommodation constraints and the thermal control, the active lens is magnified by two confocal paraboloidal reflectors. To validate the design procedure, experimental results are compared with numerical ones.

**Keywords:** active antennas; discrete lens antennas; transmit (Tx) and receive (Rx) antennas; confocal antennas; multibeam antennas; Ku-band satellite antennas





## 1. Introduction

Discrete lens antennas have recently been considered for multibeam space applications with promising results [1–10]. They are also known as constrained or bootlace lenses. Three-dimensional discrete lens antennas guarantee a large field of view, absence of blockage, double polarization capabilities, and true time delay behavior. In addition, their complexity increases slowly with the number of beams. More conventional and consolidated beamforming networks, such as the Butler matrices [11], are intrinsically lossless but do not permit the generation of a high number of beams (i.e., 500 or 1000) and do not guarantee a true time delay. If active elements are included within the lens, the low power spillover generated by the primary feed array becomes negligible and the antenna, with a single main aperture, is able to generate a multiple beam coverage with good isolation between iso-color beams [3,4,6]. The main criticalities of discrete lens antennas are associated with their mass, physical accommodation, power consumption, and thermal control of the high-power amplifiers. Reflector antennas exhibit some complementarity concerning lens antennas: they guarantee a high gain but limited scanning capabilities. Combining a discrete lens with a reflector system allows reducing the physical size of the lens while reducing the limitations of reflector antenna systems. The integration of a discrete lens with an imaging reflector system permits the generation of a multibeam coverage with narrow spot beams using a single aperture and reducing the dimensions of the lens. At the same time, if the discrete lens has the Tx and Rx capabilities, the overall system can work in transmitting and receiving modes using a single antenna aperture. Another possibility to reduce the mass and volume of the lens architecture includes increasing the operational frequency.

An imaging technique based on a single reflector allows the generation of high-directivity beams, but coma aberrations, commonly associated with the off-axis properties

of a paraboloid, and spillover present strong limitations [12]. For these reasons, imaging systems based on a single reflector are not considered in this development. An imaging optical system based on two reflectors can mitigate both drawbacks [13–16]. In its simpler architecture, the Gregorian configuration, the optics consist of the main reflector and a sub-reflector which are two confocal paraboloids (see Figure 1). Despite the increased volume compared to architectures based on a single reflector, this dual reflector configuration guarantees improved performance.

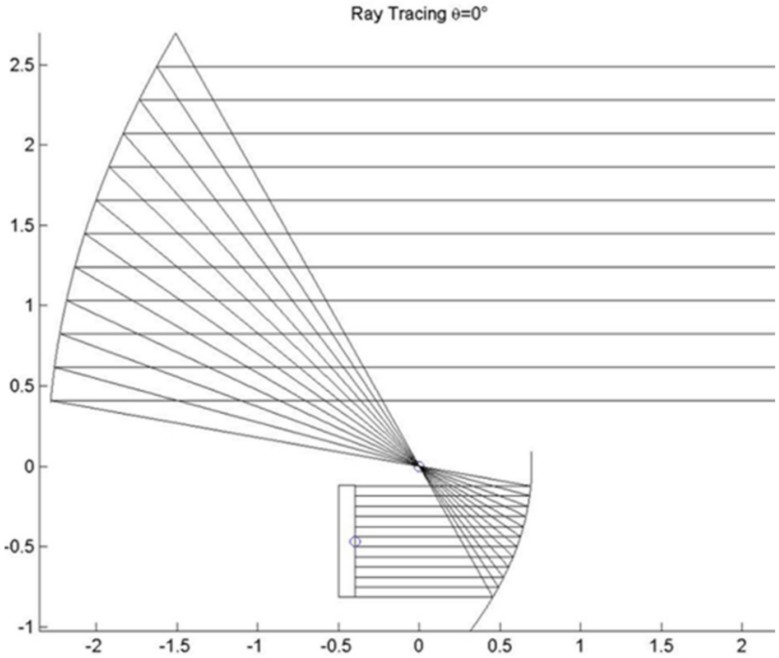

**Figure 1.** Gregorian arrangement of two confocal paraboloids magnifying a small array.

A multibeam antenna based on the integration of a passive discrete lens with a reflector system has the great advantage of being used both in transmission and in reception, requiring only a single radiating aperture for both operation modes. Furthermore, a passive discrete lens is much simpler and lighter than an active one because neither amplifiers nor a thermal control system is required within the lens itself. This greatly simplifies the accommodation, onboard the satellite, of the entire antenna system. Unfortunately, all these benefits are jeopardized by the spillover loss characterizing passive single-feed-per-beam (SFPB) architectures using a single main aperture. This is the reason why conventional SFPB antenna systems are based on three or four main apertures. The poor illumination efficiency is an intrinsic problem for passive SBPF systems based on either reflectors or discrete lenses, with the only difference that discrete lenses involve additional losses: the aperture mismatching of both back and front arrays and the ohmic loss of the RF connections between the elements of these two arrays.

An active lens is able to minimize and compensate all the losses from the primary feed up to the solid-state power amplifiers (SSPAs) and low-noise amplifiers (LNAs) inputs. The spillover loss, the mismatching loss on the back array aperture, and the ohmic losses due to the RF connections inside the lens become negligible effects because of the amplification implemented at the inputs of the front array elements. Consequently, it becomes possible to use lens configurations that provide excellent scanning properties at the expense of the low efficiency of the back aperture lens array. One of the simplest three-dimensional discrete lens architectures was proposed by Mc Grath [17].

In this paper, the design of a magnified active Tx/Rx multibeam antenna based on a discrete constrained lens is proposed for the first time. The selected operational frequency is the Ku band, a frequency that guarantees a limited separation between the transmitting

and receiving frequencies. The key element for the development of this novel antenna configuration is represented by an RF chain able to work in the Tx and Rx Ku band. In the following sections, a detailed description of a dual-band radiating element is considered.

Our goal is to give a full overview of the antenna system, a detailed description of the main subsystems, and the measurement of the key elements and technologies which enable the main functionalities. For this scope, the antenna mission scenario is first defined, then the overall antenna system is described, and finally, the lens and the radiating elements are deeply investigated.

## 2. Design of the Active Antenna

The reflectors, in confocal configuration, are arranged so that a virtual magnified image of the array is formed in front of the aperture of the main reflector [13–16]. The magnification factor, M, relating the diameters of the main reflector and the illuminating array, represents the ratio between the focal distances of the main and sub paraboloidal reflectors. M is usually larger than 2, most frequently around the value 3, but can be larger values for specific applications with reduced scanning requirements. In practice, the array is much smaller than the main reflector. This configuration, in a limited scanning region, is free from coma lobes while the spillover loss is efficiently limited with a sub-reflector oversizing, as there is little, if any, advantage in increasing the main reflector size. These advantages are offset by a limited scanning capability associated with this antenna system. The scan range in all planes is inversely proportional to the magnification factor: in practice, the number of beamwidths scanned is constant and can only be increased by adjusting the ratio between the reflector and sub-reflector focal lengths. The magnification property enables the use of a very small active lens reducing the complexity and maintaining the spot beam dimension and the overall antenna gain.

The ideal scenario, for the onboard antenna working in the Tx and Rx Ku band, is a regional multibeam coverage with 30 active beams based on a four-color reuse scheme, see Figure 2. Each spot beam has a diameter of 0.85°, the EIRP per beam is 61 dBW, and the DC power consumption is in the 6 kW range. The antenna system operates in the Rx frequency range of 14–14.5 GHz and the Tx frequency range 11.7–12.2 GHz. The antenna system is designed to work in dual linear polarization.

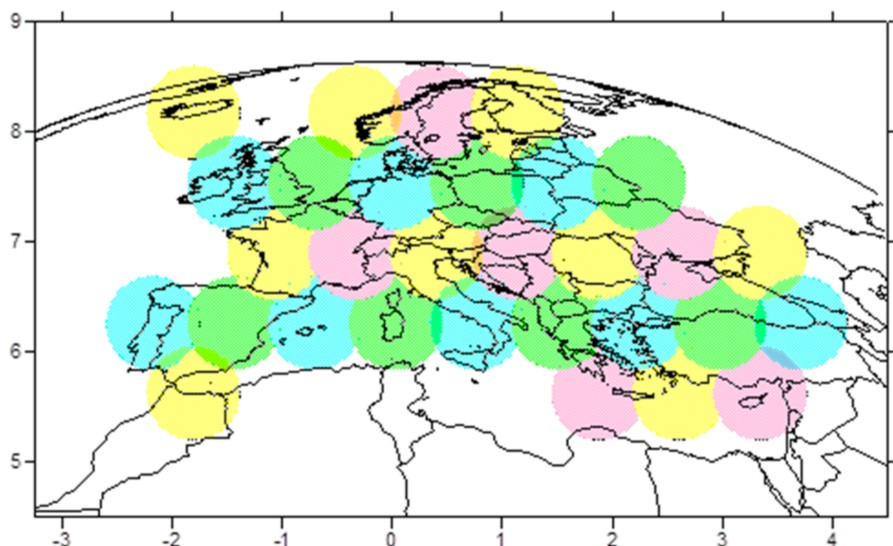

**Figure 2.** European multibeam coverage.

The classical Mc Grath discrete lens with both planar arrays, modified by repointing the back array elements (whose element apertures change size with their radial position to create an optimal field distribution), is particularly advantageous for the design of an active lens to be integrated with a reflector system.

A three-dimensional discrete lens possesses five degrees of freedom. By enforcing the planarity of the front lens and the back lens, two degrees of freedom are lost. An additional degree of freedom is spent ensuring that the elements in the front lens are radially aligned as compared to the homologous elements in the back lens. With two degrees of freedom remaining, the McGrath lens then has two perfect focal points which are collocated at the same point in the lens axis. When enforcing the two equi-length path geometrical optics conditions associated with two focal points, the two unknowns, i.e., the radial position of the back array elements with respect to the front array elements, and the length of the transmission lines connecting each couple of elements in the two arrays can be analytically derived.

Other three-dimensional discrete lens configurations could be considered, for instance, the ones proposed in [18–20]. In these lens configurations, the back profile is concave instead of flat. The resulting optimum focal distances are shorter as compared to the ones obtained for the McGrath solution. This means that more compact lens architectures can be obtained. However, this increased compactness is traded off when compared to the simplicity of the McGrath solution based on two flat profiles.

As the active lens has to be used together with a dual-reflector imaging system, the lens scan range must be M-times larger compared to the overall antenna system. As a rule of thumb, scanning the lens beam at a certain angle corresponds to a tilt in the pattern of the entire antenna roughly M times smaller and in the opposite elevation direction.

Therefore, to obtain a very large scan range, it is necessary to use very small radiating elements whose minimum physical sizes are mainly constrained by the dimensions of the active modules. The active modules comprise the septum polarizer, diplexers, and the active components required to amplify the Tx and Rx signals of the two orthogonal polarizations at the ports of the front array elements.

The design starts from the radiating aperture containing the active modules. To avoid an increased complexity of the lens, the front array is designed with a number of active elements of around 500–600. Figure 3 shows, in blue, the 29 mm$^2$ apertures of a front array with 593 elements. The complete array fits in a square with 794 mm sides. A square Cartesian lattice is chosen to overcome some mechanical and thermal issues. This type of lattice permits the simplifying of the cable connections between the back and front array and HPs position, while the periodicity of the array allows the maximum performance in terms of gain and beam scanning. The dimension of the horns filling the front array, 29 mm, represents the maximum dimension to maintain the scanning performance of the lens and to guarantee the use of the most appropriate manufacturing methods for the Tx/Rx active modules.

To control the side-lobe level, a multi-step aperture distribution on the front array was created. In particular, a front array with a two-stepped distribution was designed. It implies using two different power ratings for the SSPAs to minimize the efficiency losses of the power amplifiers. To feed the SSPAs with two different power ratings, an optimization of the external elements of the back array (see red squares in Figure 3) is performed. Having changed the dimension of the back elements, it is possible to feed the front array elements with different levels of power while maintaining the front array geometry [5–10]. This technique avoids the use of a passive attenuator device (PAD) and therefore the RF chain does not have any additional DC consumption.

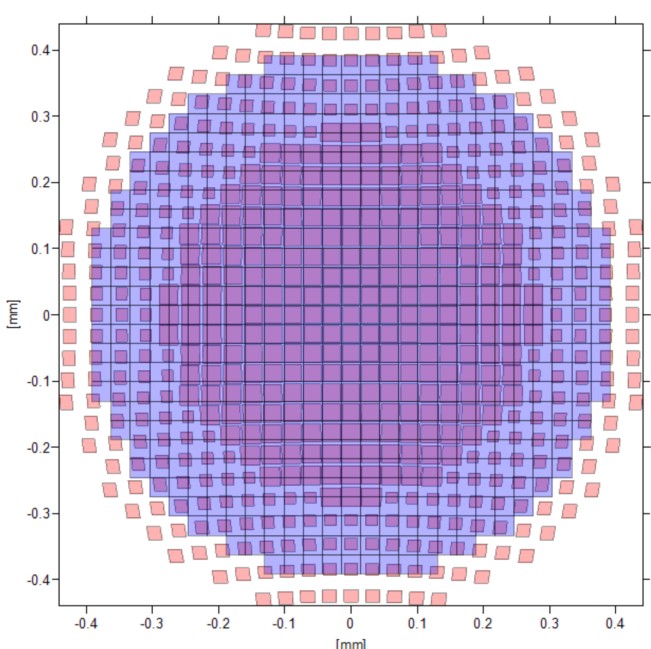

**Figure 3.** In blue, the 593 elements front array with 29 mm square apertures. In red, the optimized 593 elements of the back array.

As described in the previous sections, the Cartesian lattice solves some mechanical issues on the accommodation of the heat pipes. This simple configuration guarantees some advantage in terms of electrical performance as well. The fully populated square lattice gives a high gain, high antenna aperture efficiency, and good scanning performance compared with the typical lattice used for a phased array.

The assembly of the back and front array is reported in Figure 4, where the heat pipes of the thermal control system are located behind the active elements to extract the dissipated heat.

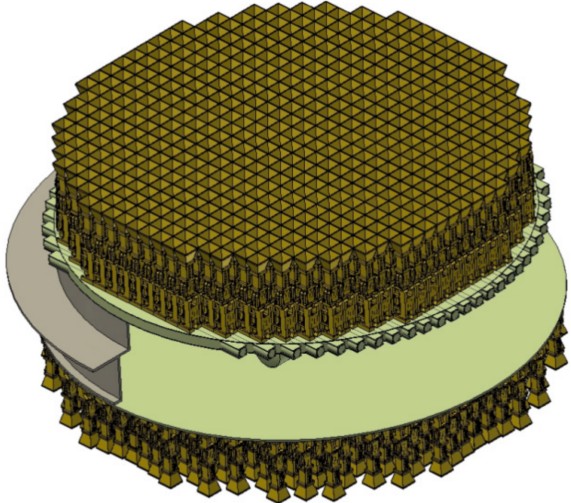

**Figure 4.** Front and back array assembly.

The last sub-assembly of the lens is the primary feed cluster. This cluster of radiating elements is located on the focal plane. For each element of the primary cluster, a well-defined beam is radiated by the discrete lens. The primary feed illuminates the back array which collects the signal and transfers it, with the proper phase and amplitude tapering, to the elements of the front array. The position of each element of the primary feed cluster

depends on the beam in the Earth's coverage: there is a single feed per beam antenna. For each beam in the coverage, a suitable feed element in the primary cluster is required. It is important to note that the discrete lens can be considered a simplified BFN where most of the beamforming is implemented in the free space, and part is implemented through simple one-to-one connection lines between the back and the front lens.

According to the active module dimensions, the complete lens front array fits inside a square with 794 mm sides. Due to the desired 0.85° spot beam diameter, the antenna reflector aperture has a diameter of 2.32 m. It follows that the magnification factor (M) of the imaging dual-reflector system must be 2.32/0.794 = 2.92. Additionally, as the Ku-band multibeam coverage extends on an angular sector of ±3° (see Figure 2), the scanning region of the discrete lens must be 3·2.92~9° due to the magnification factor. The focal of the main reflector is 2.784 m while the sub-reflector has a diameter of 1.307 m and a focal of 0.98 m. The sub-reflector is partially oversized to reduce the effect of the spill-over due to the lens illumination. The distance between the lens antenna and sub-reflector, which is 1.470 m, is calculated according to the antenna design guidelines provided by Dragone [14], using the formula:

$$C_1 B_1 = |F_1 B_1| \frac{M+1}{M}, \tag{1}$$

Figure 5 depicts the antenna system constituted by the dual reflectors and the discrete lens.

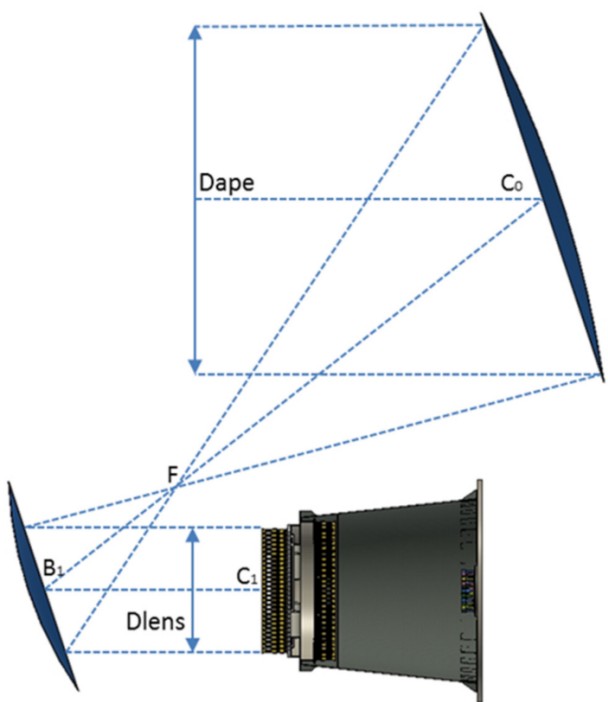

**Figure 5.** Discrete lens and confocal dual reflector system.

The scanning properties of the active lens combined with the reflector over the entire coverage region are shown in Figures 6 and 7 for the central frequency of the Tx and Rx band and the azimuth plane ($\varphi = 0$ degrees). The different scanned beams are obtained by feeding the active antenna with the corresponding element in the primary cluster array.

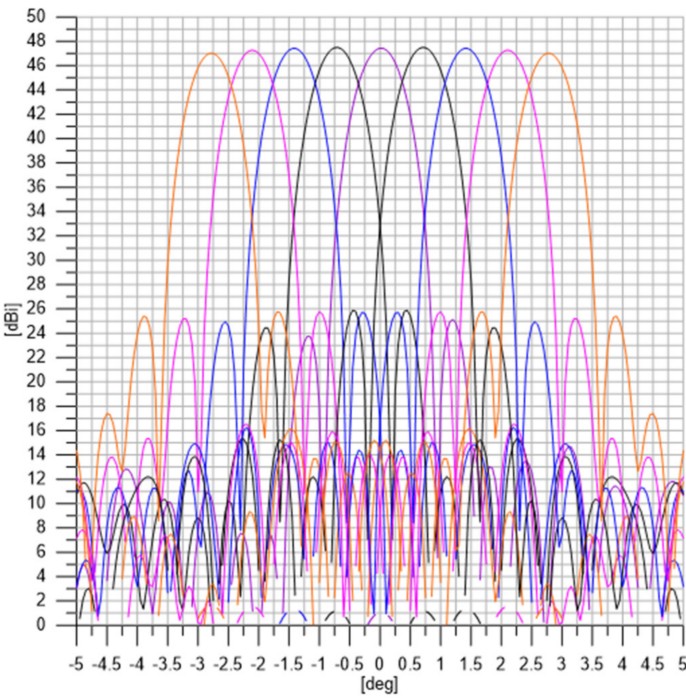

**Figure 6.** Radiation pattern and scanning performance in the Tx frequency band.

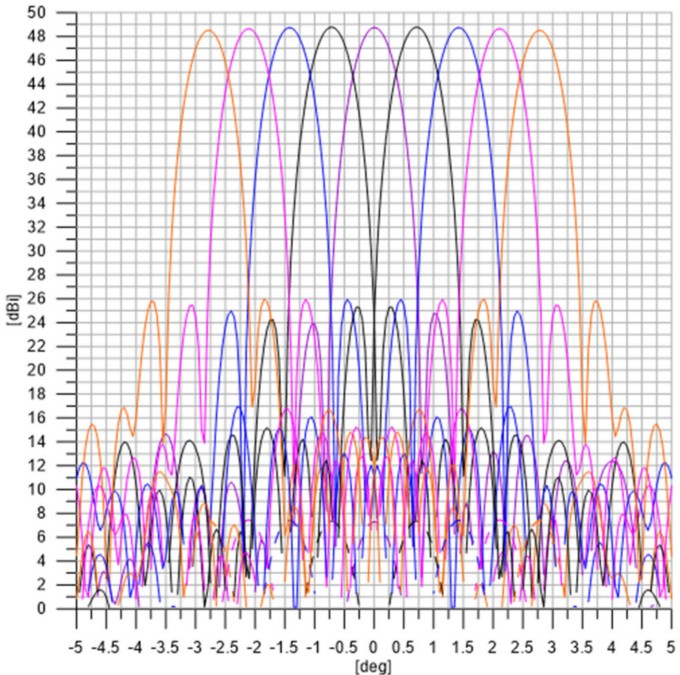

**Figure 7.** Radiation pattern and scanning performance in the Rx frequency band.

Figure 8 shows that the maximum scanning losses of the antenna are around 0.75 dB and the sidelobes level (in Figure 7) are in line with the desired values in the scanning range of ±3° for the Tx and Rx band.

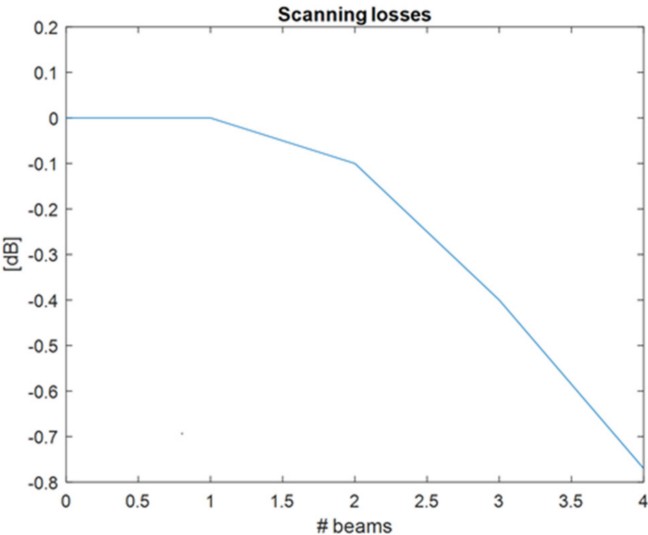

**Figure 8.** Antenna scanning losses.

### 3. Design of the Tx/Rx Radiating Element

As described in the previous section, the novelty of the proposed design is mainly associated with the single antenna aperture guaranteeing the multibeam coverage in the Tx and Rx Ku band. This requires radiating elements that are able to cover the dual-frequency band. Due to the discrete lens antenna configuration, two different arrays were designed, the front and the back array. For each element of the front array, an element on the back array is needed. In the Tx path, the elements of the back array collect the signal from the primary feed and transfer it to the front array that is responsible for radiating the desired beam toward the Earth's surface. In the Rx path, the elements of the front array collect the incoming signal from the Earth and bring it to the back array, then the back array transmits it to the primary feed.

An important aspect for the design of the arrays is related to the design of the radiating elements, which have a key role in the final antenna performance. It is necessary to choose the optimal configuration of the radiating element to reduce their dimensions and enable the dual-band functionalities. Figure 9 reports the building block of the RF chain inside the discrete lens.

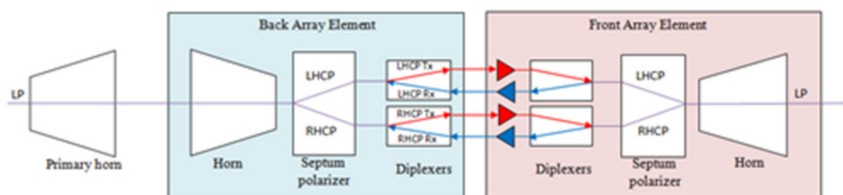

**Figure 9.** RF chain of the active discrete lens.

Both the back and the front array elements are composed of horns, septum polarizers, and diplexers. This assembly is called a module. The only difference between the elements of these two arrays is that the front array elements are equipped with the active components (active module), SSPA in the Tx paths, and LNA in the Rx paths.

According to the requirements of the Ku-band multibeam coverage, reported in Section 2, the active lens has to operate in the two frequency bands, 11.7–12.2 GHz and 14.0–14.5 GHz, using the two linear polarizations (HP and VP). Figure 9 shows a schematic representation of the Tx/Rx module and the relative polarization transduction within the active lens which allows the conversion of the linear polarizations into two opposite circular polarizations. Considering the Tx case, and assuming the primary horn receives

a linear polarization (i.e., a vertical or horizontal one), the back horn receives this field, which is split into two signals with two opposite circular polarizations using a septum polarizer. The signals follow the Tx paths in the diplexers and then, by means of RF cables, are delivered to the front array. Here, after the power amplification, the Tx signals reach the diplexers, and then a septum polarizer reconstructs the original linear polarization summing up the two opposite circular polarizations.

This polarization transduction is necessary in order to avoid the use of a conventional orthomode transducer (OMT) operating in orthogonal planes, which would not permit the obtaining of radiating chains with a compact footprint. Conventional OMTs have perpendicular waveguide inputs, making them difficult to integrate into the front and back array, while a septum polarizer has the advantage that the waveguide inputs are perfectly parallel and array elements can be placed side by side allowing for compactness. The septum polarizers are used only for managing the linear polarization transmitted or received by the lens. The septum polarizers avoid the accommodation of other heavy devices working in linear polarization. This design solution allows for management of the two linear polarizations in the Tx and Rx path, giving the maximum flexibility in terms of polarization.

Each diplexer has two rectangular waveguides for the Tx and Rx paths of one specific circular polarization. Irises are used to filter the undesired components in the two different branches. The number of irises controls the order of the filters and the relative isolation between the Tx and Rx path.

Figure 10 shows a 3D model of the Ku-band Tx/Rx module for the front and back array, while Figure 11 depicts the detailed 3D model of the three-step septum polarizer and diplexers.

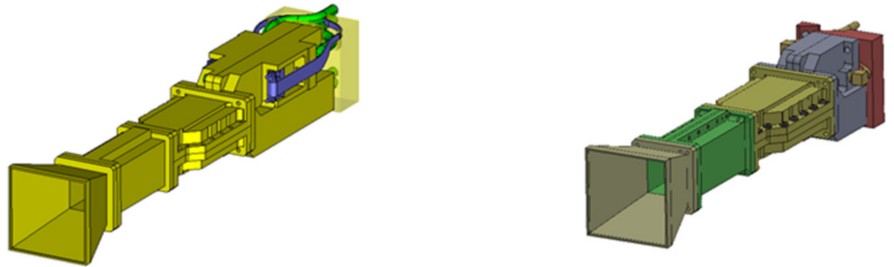

**Figure 10.** Active front element (**left**), passive back element (**right**).

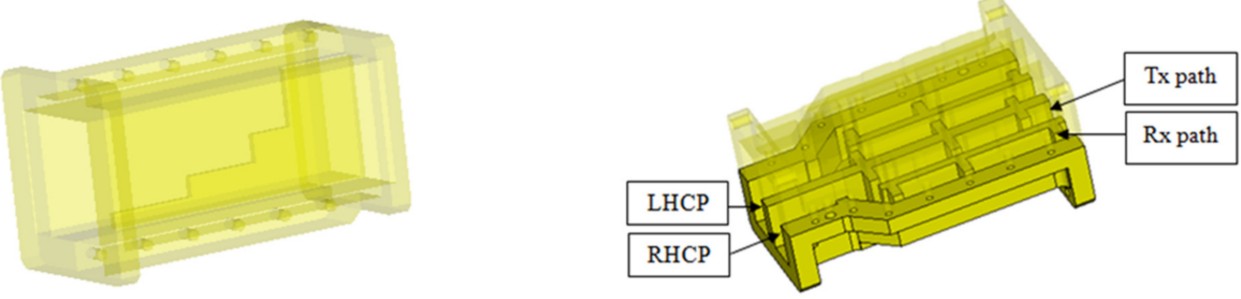

**Figure 11.** Optimized septum polarizer and diplexers for the Tx/Rx Ku-band module.

Figure 12 shows the simulated return loss for the Tx and Rx paths of the full radiating module. The S11 of −20 dB is guaranteed in the Tx and Rx frequency band while the isolation between the Tx/Rx path is larger than 56 dB, as reported in Figure 13. If requested by the system analysis, additional isolation between the Tx/Rx path can be achieved by optimizing the order of the filters in the diplexers.

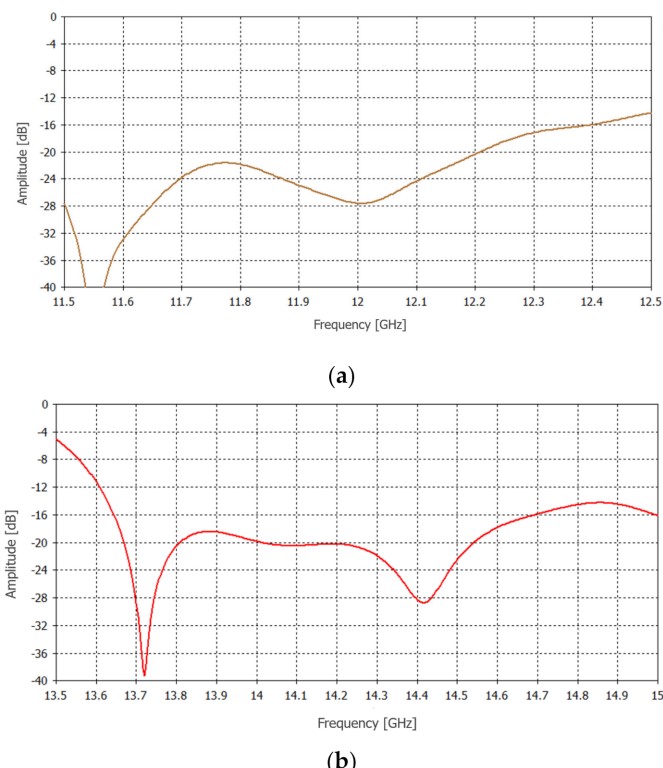

(**a**)

(**b**)

**Figure 12.** S11 for the Tx path (**a**) and Rx path (**b**).

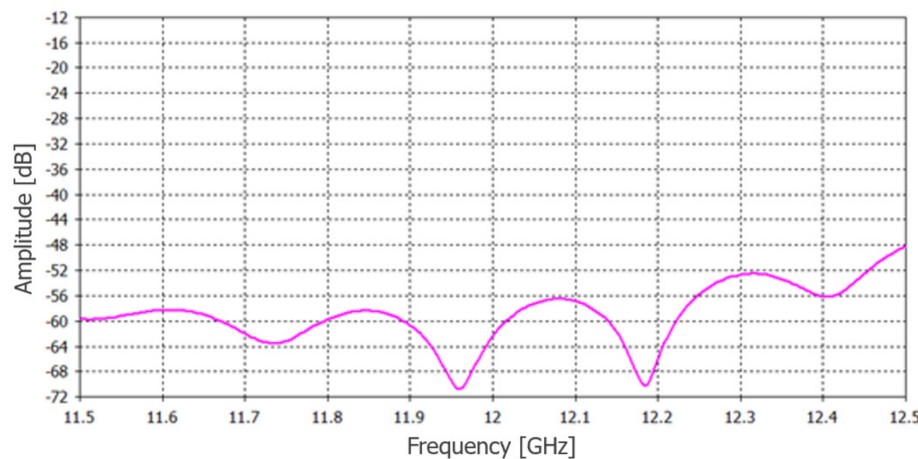

**Figure 13.** Isolation between the Tx/Rx path.

## 4. Breadboarding and Testing

Breadboarding and other relative tests aimed to prove the main electrical aspects of the Tx/Rx modules and the radiation pattern of the discrete lens. The tests were performed on the following subsystems:

- The Tx/Rx passive modules;
- The small discrete lens composed of 3 × 3 passive elements.

Section 4.1 is focused on the manufacturing and test of the Tx/Rx passive module, while Section 4.2 is focused on the radiation pattern of a small passive lens.

### 4.1. Tx/Rx Passive Module Manufacturing and Test

The scope of the Tx/Rx passive module demonstrator was used to verify the design of the single radiating element that works in the Tx and Rx Ku band. The electrical tests

were performed to verify the S parameters and the radiation performance. Figure 14 shows an example of the passive module which is fully representative of the back array element. As discussed above, the single module operates in circular polarization in order to perform the polarization transduction described in Section 3.

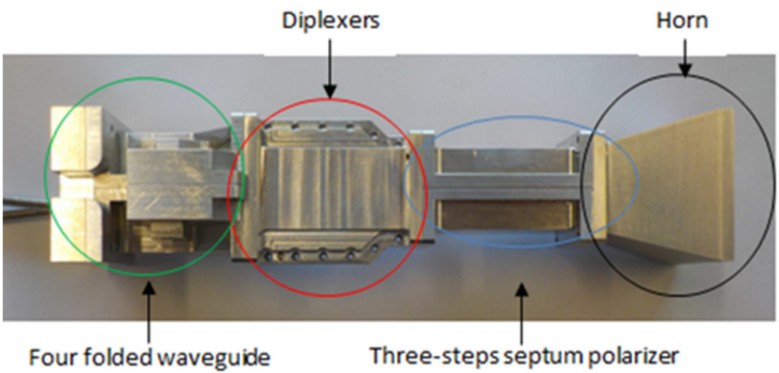

**Figure 14.** Manufactured Tx/Rx passive module.

The module has a square horn that is directly connected with a three-step septum polarizer. The septum polarizer is a two-port device. A diplexer that separates the Tx and Rx paths is connected at each port. The diplexer assembly is a four-port device with a folded waveguide connected at each port to spill the signal and to feed the corresponding element in the front array. Figures 15 and 16 report the measured S11 parameters of the manufactured 18 Tx/Rx modules for the two sub-bands, while Figure 17 shows the isolation between the Tx and Rx path. The measured data are in perfect agreement with the simulated S parameters (see Section 3).

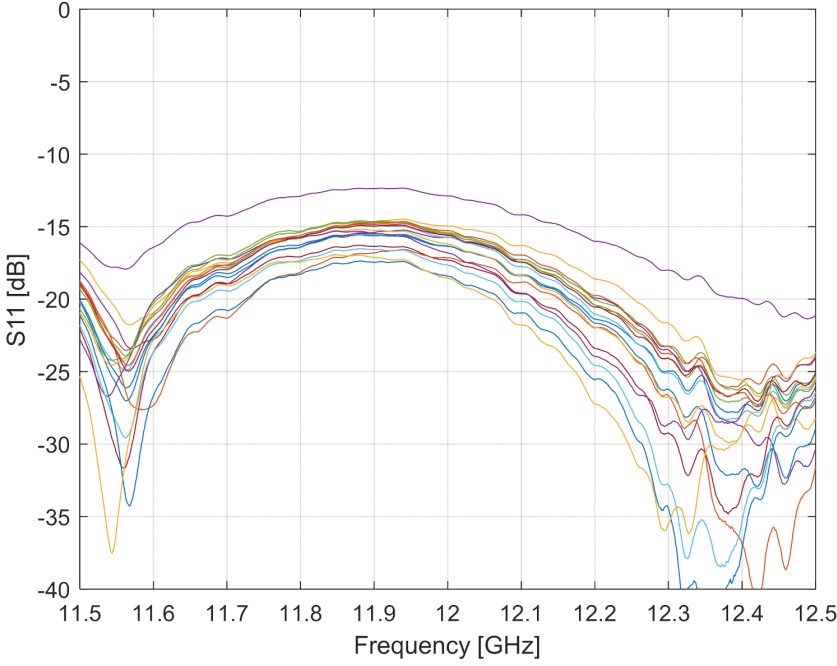

**Figure 15.** Measured S11 parameter in the Tx band.

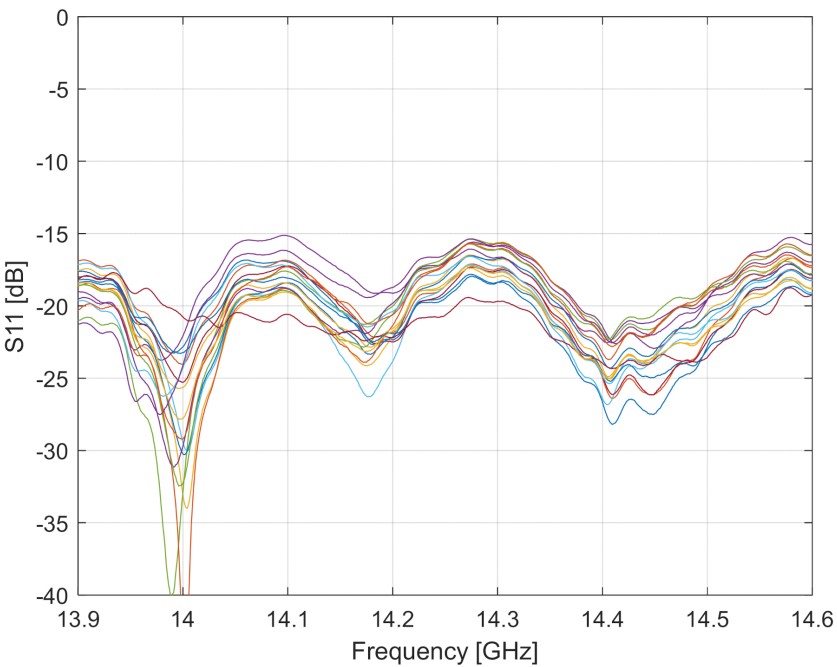

**Figure 16.** Measured S11 parameter in the Rx band.

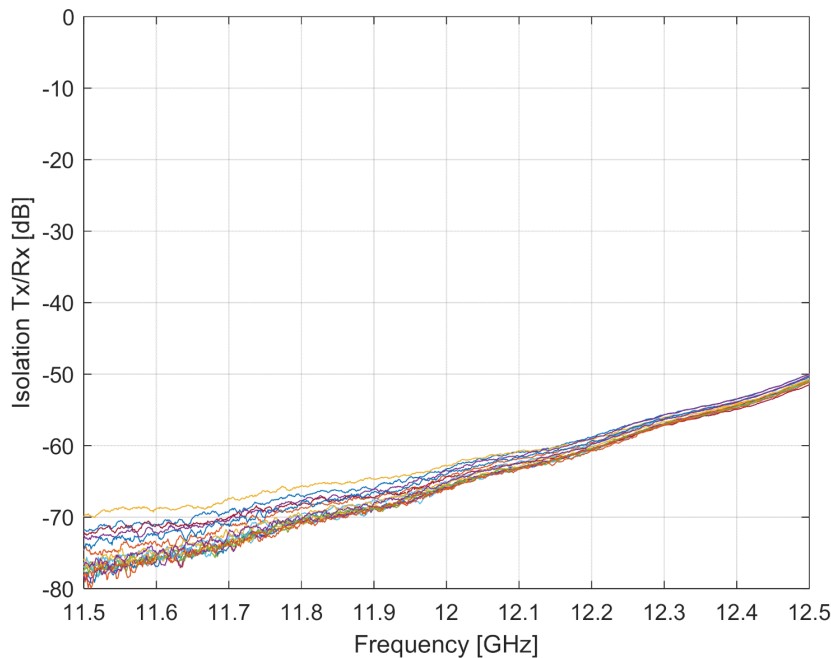

**Figure 17.** Isolation between the Tx and Rx ports.

Figure 18 depicts the radiation patterns of the Tx/Rx Ku-band module at the frequencies of 11.95 GHz and 14.25 GHz. In particular, the azimuth, elevation, and 45 degree slant angle cuts of the theoretical and measured radiation patterns are reported. The cross-polarization level is sufficiently low: this finding demonstrates that the polarization transduction inside the lens RF chain is carried out properly. The theoretical patterns are achieved by utilizing the full-wave simulation of the module using CST MWV software.

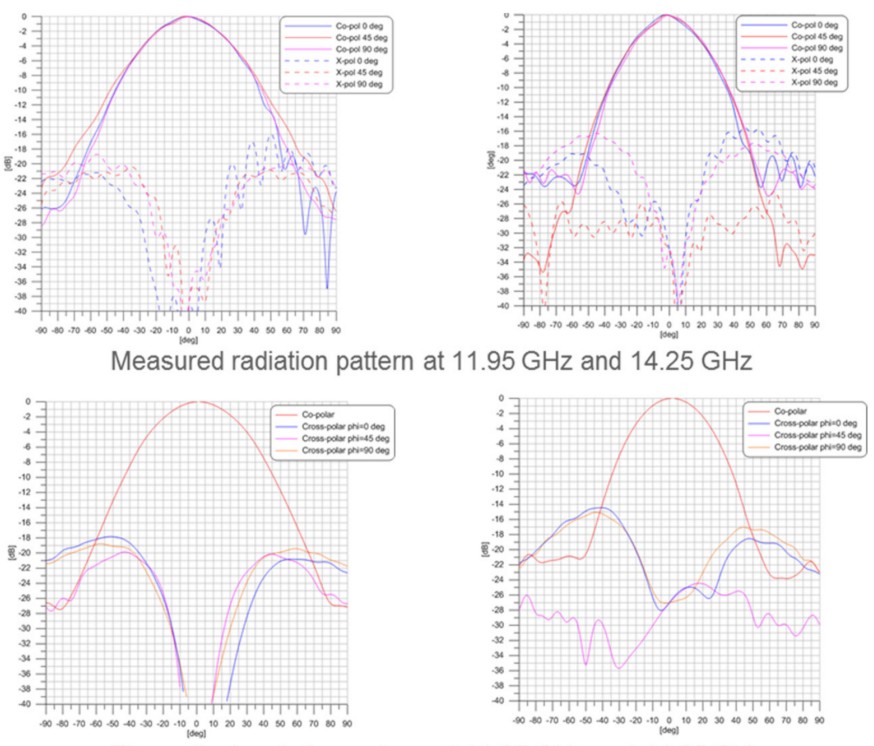

Measured radiation pattern at 11.95 GHz and 14.25 GHz

Theoretical radiation pattern at 11.95 GHz and 14.25 GHz

**Figure 18.** Comparison between the measured and theoretical pattern of the Tx and Rx module.

### 4.2. Tx/Rx Passive Discrete Lens Manufacturing and Test

After the tests on each of the 18 Tx/Rx were carried out, the elements were arranged to realize a small discrete lens with 3 × 3 elements for the front and back array as shown in Figure 19. This small lens is representative of the central part of the fully designed one.

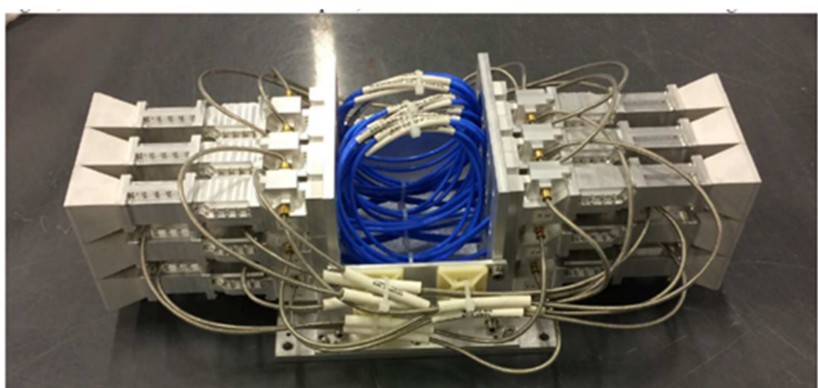

**Figure 19.** Assembled Tx/Rx discrete lens.

Due to the small dimension of the lens, the only test that could be performed was the one related to the measurement of the radiation pattern concerning the beam pointing at 0°. Any other kind of scanned beam cannot be measured because the primary feed moves in the focal plane and the field detected by the reduced lens is smaller than the spillover.

The normalized measured radiation pattern for the Tx and Rx frequencies is shown in Figure 20. Table 1 reports the comparison between the measured and theoretical gain of the 3 × 3 discrete lens. There is a slight difference of 0.18 dB in the Tx and Rx frequency band.

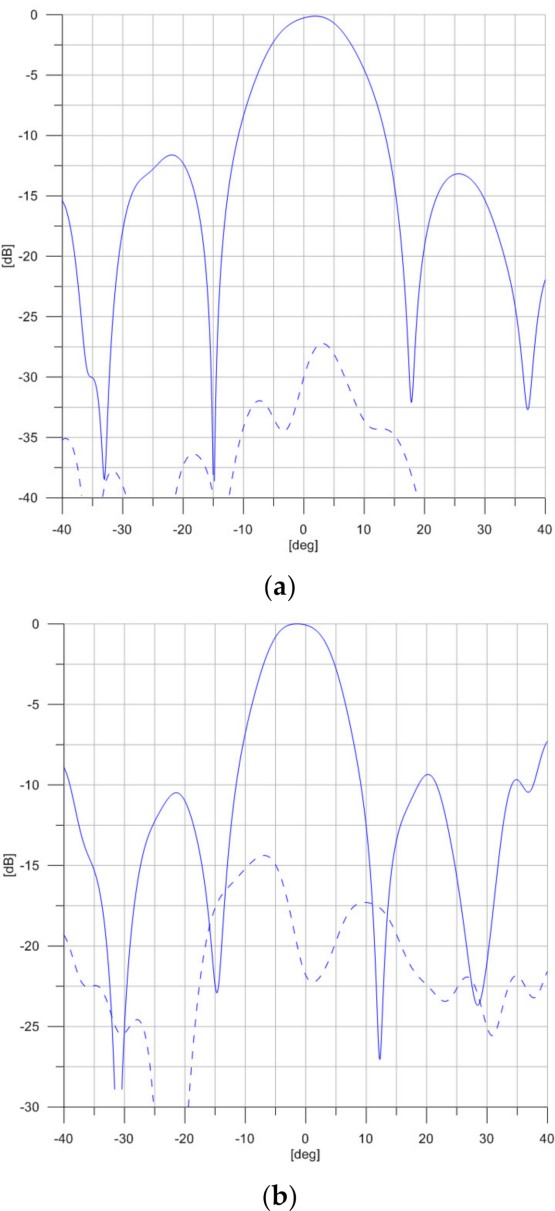

**Figure 20.** Tx (**a**) and Rx (**b**) radiation pattern of the small Tx/Rx lens.

**Table 1.** Measured and theoretical gain for the $3 \times 3$ discrete lens.

| Frequency (GHz) | Theoretical Gain (dBi) | Measured Gain (dBi) |
|:---:|:---:|:---:|
| 11.95 | 20.82 | 20.64 |
| 14.25 | 22.23 | 22.05 |

## 5. Conclusions

A transmit-receive Ku band antenna system comprising a feeding array, a discrete active lens, and a dual confocal reflector was developed. A radiating element working in Tx and Rx in two opposite linear polarizations was proposed. After the definition of the radiating elements, the active lens was derived in consideration of the integration with a confocal imaging system to generate a multibeam coverage with spot beams characterized by a 0.85° beam width. A periodic front array in the lens configuration was chosen in order to optimize the heat dissipation via heat pipes (HPs) and to keep a simple feeding network for the active elements. To reduce the side-lobe level, a two-step distribution was created

thus modifying the dimension of the elements of the back array. The integration of the active discrete lens with a reflector system permitted the reduction in the dimension of the active lens with a consequent reduction of the overall complexity of the radiating system. The Tx/Rx radiating elements were manufactured and tested. The agreement between the simulated and measured data proved the effectiveness of the design procedure. Moreover, the radiating element was arranged to realize a $3 \times 3$ discrete lens able to test a beam pointing at the boresight.

**Author Contributions:** Data curation, G.T.; Formal analysis, P.G.N.; Investigation, G.R., P.G.N. and G.T.; Methodology, G.R. and P.G.N.; Software, G.R. and P.G.N.; Supervision, G.T.; Validation, G.R.; Writing—original draft, G.R., P.G.N. and G.T.; Writing—review & editing, G.R., P.G.N. and G.T. All authors have read and agreed to the published version of the manuscript.

**Funding:** This research was funded by ESA ARTES activity "Discrete Lens for Array Fed Reflector Antennas" (contract No 4000111762).

**Acknowledgments:** The authors G. Ruggerini and P. Nicolaci used to work in Space Engineering S.p.A. (now Airbus Italia) when involved in this activity.

**Conflicts of Interest:** The authors declare no conflict of interest.

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
