# Peer review of "A Ku-Band Magnified Active Tx/Rx Multibeam Antenna Based on a Discrete Constrained Lens"

_electronics, doi:10.3390/electronics10222824_

Round 1

Reviewer 1 Report

The paper presents the design, manufacturing and test of a Ku-band magnified active Tx/Rx multibeam antenna based on a discrete constrained lens.

The work is interesting, well written and well organized. However some things should be cleared and have to be improved:

  • Fig 6-7, how did you obtained these radiation patterns? Which planes are you referring to?
  • line 204: "Fig. 8 shows"
  • line 209 "able to perform"
  • line 219 "in fact"
  • Fig. 13: i know that in the y-axis you are talking about dB, but put it in the scale.
  • Figs. 15-16-17, they are very low quality images, please improve them
  • Fig. 18, very low quality image, unreadable axis. How did you obtain these radiation patterns (clearly, I'm not referring to the measured patterns);
  • Fig. 20, low quality image with unreadable axis.
  • why didn't you realize a larger prototype and just a 3x3 array?

Author Response

We thank the Reviewer #1 for her/his suggestions. A feedback is proposed here.

 Fig 6-7, how did you obtained these radiation patterns? Which planes are you referring to?

The radiation mechanism is explained in rows 177-187. The active antenna lens has a primary feed cluster, one feed for each beam of the coverage. Feeding the antenna with a primary feed corresponds to radiate a single beam in the coverage.

The simulations are performed following several steps. First of all, the radiation pattern of each primary feed is calculated by means of the full wave simulation in CST MWV. These patterns are expressed in a Spherical Wave Expansion (SWE) format. The same procedure is used for the elements of the back and front array. Then, an in-house software based on the FRIIS vectorial formula is used to calculate the power delivered to the front array. The front array and the reflector system are simulated adopting the commercial tool GRASP (by TICRA) in order to obtain the final radiation pattern.

Added rows 202-204 for further explanation describing the plane for the radiation pattern 

  • line 204: "Fig. 8 shows"

Corrected

  • line 209 "able to perform"

Corrected

  • line 219 "in fact"

Corrected

  • 13: i know that in the y-axis you are talking about dB, but put it in the scale.

New figure edited with all the scales, see rows 279-280.

  • 15-16-17, they are very low quality images, please improve them

New figures have been edited, see rows 306-312.

  • 18, very low quality image, unreadable axis. How did you obtain these radiation patterns (clearly, I'm not referring to the measured patterns);

The figure has been modified as requested. The theoretical patterns are achieved by means of the full wave simulation of the module using CST MWV software. See rows 218-219.

  • 20, low quality image with unreadable axis.

The figures have been modified, see rows 336-340

  • why didn't you realize a larger prototype and just a 3x3 array?

The dimension of the active lens antenna is limited to 3x3 array due to the limited budget available in the activity. In other activities we have tested a full lens antenna system (see reference [1], [2] and [4]). The main novelty in this activity is the dual band achieved adopting the new module in the front and back array. The limited lens allows anyway to evaluate how the polarization transduction works in the full RF path.

Reviewer 2 Report

A Ku-band magnified active 10 Tx/Rx multibeam antenna based on a discrete constrained lens concept is designed, manufactured and measured. The proposed antennas can generate multi-beams over the working frequency bandwidth and work in dual-polarization radiation. The combination between its transmitting and receiving functionality of this active antenna is realized. The issues of accommodation constraints and thermal control are solved well by using two confocal paraboloidal reflectors. The experimental results are good matched well with the designed ones. This research is of great significance in satellite communication.

Author Response

We thank the Reviewer #2 for her/his feedback.

Reviewer 3 Report

This paper presents the design, manufacturing and test of a Ku-band magnified active Tx/Tx multibeam antenna based on a discrete constrained lens. The work is robust and the antenna shows advantageous performances. The writing of this paper should be improved. Most of the figures are not clear enough. For some figures, the horizontal and vertical quantities are not provided. Please improve.

Author Response

We thank the Reviewer #3 for her/his suggestions. A feedback is proposed here.

Several figures have been improved with a better resolution and adding the missing scale, more details in the feedback to Reviewer #1.

The entire manuscript has been further polished and improved.

Reviewer 4 Report

Please add the necessary design details of the front/back array/lattice. 

Please explain the scanning mechanism. 

Please explain how the multibeam is formed or distributed.

What is the measured gain?

Please improve the quality of many figures in the manuscript (e.g. unreadable texts in the figure).

Please proofread and further polish the language, as there are quite some typos and grammar errors in the manuscript.

Author Response

We thank the Reviewer #4 for her/his suggestions. A feedback is proposed here.

The following lines have been added:

“As described in the previous sections, the Cartesian lattice allows to solve some mechanical issues on the accommodation of the heat pipes. This simple configuration guarantees as well some advantage in terms of electrical performance. In fact, the fully populated square lattice gives a high gain, high antenna aperture efficiency, and good scanning performance compared with typical lattice used for phased array”. See rows 165-169

Please explain the scanning mechanism. 

Added the following section (see rows 176-185)

The last sub assembly of the lens is the primary feed cluster. This cluster of radiating elements is accommodated on the focal plane. For each element of the primary cluster a well-defined beam is radiated by the discrete lens. In fact, the primary feed illuminates the back array which collects the signal and transfer it, with a proper phase and amplitude tapering, to the elements of the front array. The position of each element of the primary feed cluster depends on the beam in the Earth coverage: it is a single feed per beam antenna. For each beam in the coverage a suitable feed element in the primary cluster is required. It is important to note that the discrete lens can be considered a simplified BFN where most of the beamforming is implemented in the free-space, and a part is implemented through simple one-to-one connection lines between the back and the front lens.   

Please explain how the multibeam is formed or distributed.

Added the following section (see rows 176-185)

The last sub assembly of the lens is the primary feed cluster. This cluster of radiating elements is accommodated on the focal plane. For each element of the primary cluster a well-defined beam is radiated by the discrete lens. In fact, the primary feed illuminates the back array which collects the signal and transfer it, with a proper phase and amplitude tapering, to the elements of the front array. The position of each element of the primary feed cluster depends on the beam in the Earth coverage: it is a single feed per beam antenna. For each beam in the coverage a suitable feed element in the primary cluster is required. It is important to note that the discrete lens can be considered a simplified BFN where most of the beamforming is implemented in the free-space, and a part is implemented through simple one-to-one connection lines between the back and the front lens.     

What is the measured gain?

Added Table 1 which reports the theoretical and measured gain, (see rows 342-343)

Please improve the quality of many figures in the manuscript (e.g. unreadable texts in the figure).

New figures are provided

Please proofread and further polish the language, as there are quite some typos and grammar errors in the manuscript.

The entire manuscript has been further polished and improved.

Round 2

Reviewer 4 Report

Thank you for addressing my comments.